# Simulation of 1500 °C Thermal Shock for Novel Structural Thermal/Environmental Barrier Coatings System

**Kaibin Li** **, Weize Wang \*, Ting Yang and Yangguang Liu**

Key Laboratory of Pressure System and Safety, Ministry of Education, East China University of Science and Technology, Shanghai 200237, China
* Correspondence: wangwz@ecust.edu.cn

**Abstract:** In recent years, with the development of SiC composites in aero-engine hot-end components, environmental barrier coatings (EBCs) have received extensive attention. Moreover, in order to elevate the service temperature, it is a developing trend to apply thermal barrier coatings (TBCs) with low thermal conductivity on EBCs coating system to form thermal/environmental barrier coatings (T/EBCs). However, the combination of high coefficient of thermal expansion (CTE) of TBCs with low CTE of EBCs often leads to premature failure due to excessive thermal expansion mismatch stress. However, a novel structural thermal barrier coating with embedded micro-agglomerated particles (EMAP TBC) by using atmospheric plasma spraying (APS) process has brought hope to solve this problem due to its low elastic modulus. Therefore, in this study, an innovative EMAP $Gd_2Zr_2O_7$ T/EBCs coating system (EMAP $Gd_2Zr_2O_7/Yb_2Si_2O_7/Si$) under 1500 °C flame thermal shock was simulated and systematically studied on the SiC substrate. The results showed that the EMAP $Gd_2Zr_2O_7$ T/EBCs coating system has much lower thermal stress than the conventional $Gd_2Zr_2O_7/Yb_2Si_2O_7/Si$ T/EBCs coating system. Furthermore, when the thickness of each layer of the EMAP $Gd_2Zr_2O_7$ T/EBCs coating system varies, to meet the thermal insulation requirements of $Yb_2Si_2O_7$ layer and reduce the thermal shock stress, the thickness of the EMAP $Gd_2Zr_2O_7$ layer is recommend being about 100 μm. Meanwhile, the thicknesses of $Yb_2Si_2O_7$ and Si layers can be set as large as needed. In addition, with the increase in $Yb_2SiO_5$ doping content in the $Yb_2Si_2O_7$ intermediate layer, the EMAP $Gd_2Zr_2O_7$ T/EBCs coating system suffers a greater risk of spalling failure due to the increase in thermal stress.

**Keywords:** temperature distributions; stress distributions; thickness design; ytterbium silicate; gadolinium zirconate



## 1. Introduction

Silicon carbide-based ceramic matrix composites (SiC CMCs) are the promising candidates for high-temperature structural components of engines, such as guide vanes, fairings, rotor blades, nozzle blades, etc. SiC CMC exhibits excellent oxidation resistance in dry air by forming slowly growing silicon dioxide ($SiO_2$) flakes on the surface. However, under the high temperature, high pressure, high speed and water vapor-rich combustion environment of gas turbines, $SiO_2$ continuously reacts with high temperature water vapor to forms gaseous silicon hydroxide ($Si(OH)_4$), which volatilizes rapidly and leads to unacceptable decline [1–4]. To solve this problem, environmental barrier coatings (EBCs) are often used to protect SiC-based ceramic components.

EBCs can not only eliminate the rapid volatilization of SiC-based ceramic components in water vapor rich environments, but also inhibit the rapid oxidation of components, and reduce the bearing temperature of components. Figure 1 shows the development of EBCs coating systems [5–24]. At present, EBCs have experienced three generations. The first generation is the mullite ($3Al_2O_3 \cdot 2SiO_2$) coating system [5–9]. Although mullite matches the thermal expansion coefficient (CTE) of SiC substrate and has good corrosion resistance.

However, mullite has high $SiO_2$ activity (about 0.4) and the resistance to water vapor erosion is also weak. To enhance the water vapor erosion resistibility of the coatings, Lee et al. [7–9] tried to coat the yttria-stabilized zirconia (YSZ) layer on the mullite coating surface by using the conventional APS process. However, the CTE of YSZ is relatively high (about twice that of mullite), so the YSZ/Mullite double-layer coating system is prone to produce many "through cracks" during the thermal cycle. The second generation is $BaO$-$SrO$-$Al_2O_3$-$SiO_2$ (BSAS) system coatings, such as BSAS/Mullite/Si or BSAS/Mullite+BSAS/Si [10,11,13]. The BSAS surface layer has excellent resistance to crack propagation and low $SiO_2$ activity (about 0.1), which reduces the volatilization of the coating in the water-oxygen corrosion environments. However, when temperatures exceed 1311 °C, the BSAS system forms a glassy phase with $SiO_2$, which leads to premature failure of the coating [10,11]. The third generation is the rare earth silicate system, such as ytterbium monosilicate ($Yb_2SiO_5$) and ytterbium disilicate ($Yb_2Si_2O_7$) [14,16,18,25]. Compared with BSAS, $Yb_2SiO_5$ has lower $SiO_2$ activity, high temperature phase stability, excellent water and oxygen corrosion resistance [14], and can work at higher temperatures (~1450 °C) [26], so it has attracted wide attention. Furthermore, $Yb_2SiO_5$ is less likely to form $Si(OH)_4$ than $Yb_2Si_2O_7$ in high-temperature water vapor environment [20,27,28]. However, since the CTE of $Yb_2SiO_5$ is about twice than that of the SiC substrate, it is prone to introduce many "mud cracks" during high-temperature thermal cycling [14,16–19]. Moreover, $Yb_2SiO_5$ has less resistant to the calcium–magnesium–aluminosilicate (CMAS) corrosion than $Yb_2Si_2O_7$ [29]. For $Yb_2Si_2O_7$, it has CTE matching the SiC substrate [18,20], higher fracture toughness than $Yb_2SiO_5$ [21,30], but moderate resistance to water and oxygen corrosion [14,26]. Richards et al. [20] confirmed that after the $Yb_2Si_2O_7$ (~125 μm)/Si (~50 μm) double-layer coating system is corroded for 2000 h under thermal cycle at 110–1316 °C in the water-oxygen corrosion environment, most part of the coating still remain intact.

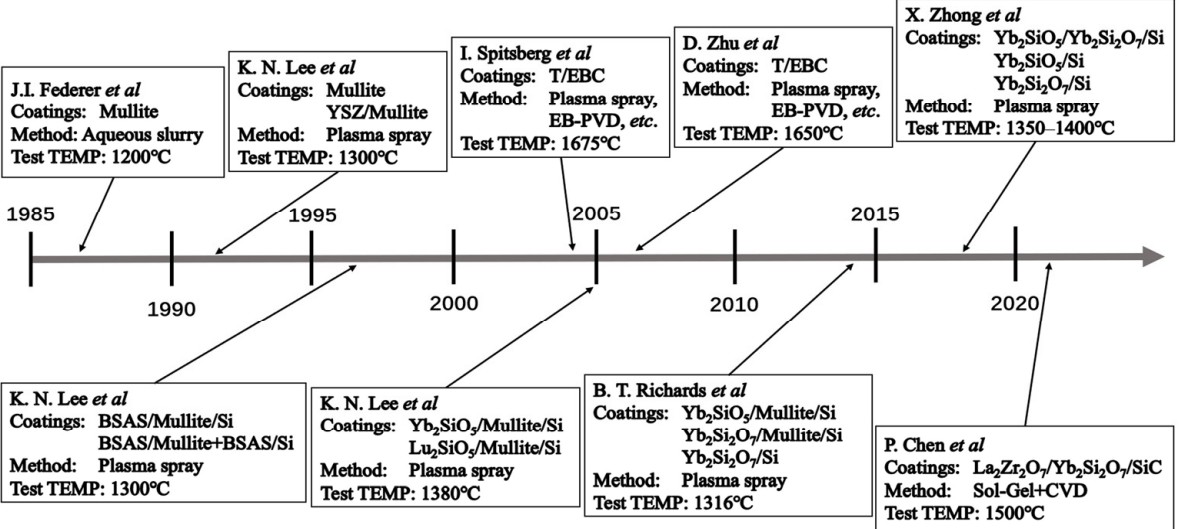

**Figure 1.** Development of EBCs coating systems [5–24].

To improve the low fracture toughness and high CTE of $Yb_2SiO_5$, the preparation of $Yb_2SiO_5$ and $Yb_2Si_2O_7$ composite coatings is a hotspot. For example, Wang et al. [30] prepared $(1 − x)Yb_2SiO_5$-$xYb_2Si_2O_7$ (x = 0, 0.1, 0.2, 0.3, 0.4, 0.5, 0.6, mol.%) composite ceramics and found that the fracture toughness of $0.5Yb_2SiO_5$-$0.5Yb_2Si_2O_7$ ceramics is more than 60% higher than that of the pure $Yb_2SiO_5$ ceramics. Garcia et al. [31] also confirmed that $50Yb_2Si_2O_7$-$50Yb_2SiO_5$ (mol.%) EBCs remain intact after 200 thermal cycles at 1200 °C. In addition, Zhong et al. [21] used $Yb_2Si_2O_7$ coating as the intermediate layer between $Yb_2SiO_5$ top layer and Si bonding layer, which also achieved good results. However, the optimum mixing ratio of $Yb_2SiO_5$ and $Yb_2Si_2O_7$ content may be different due to the different CTE combinations in different coating systems.

The service temperature of the above-mentioned EBCs is mostly below 1500 °C. To meet the needs of higher temperatures, thermal-environmental barrier coatings (T/EBCs) are implemented [32]. T/EBCs are the application of thermal barrier coatings (TBCs) on EBCs coating system. For example, Spitsberg et al. [12] and Zhu et al. [15] proposed two types of multi-layer T/EBCs successively. Both of them introduced a stress release layer/transition layer and adopted a combination process of atmospheric plasma spraying (APS) and electron beam-physical vapor deposition (EB-PVD). The prepared multi-layer T/EBCs are used for hundreds of hours and without peeling at ~1650 °C. However, the preparation process and materials used for the multi-layer T/EBCs are relatively complicated, and the coating thickness is also relatively thick (~700 μm). In addition, Chen et al. [33] pointed out that $Yb_2Si_2O_7$ layer partially melts and has a eutectic reaction with the underlying mullite layer at 1500 °C in the air after 5 h. Therefore, when the $Yb_2Si_2O_7$ layer is used at high temperature above 1500 °C, adding a TBC thermal insulation layer is highly recommended. Subsequently, Chen et al. [24] used lanthanum zirconate ($La_2Zr_2O_7$) material as the thermal insulation layer and prepared three-layer $La_2Zr_2O_7$/$Yb_2Si_2O_7$/SiC EBCs by chemical vapor deposition (CVD) and sol-gel method combined with air spraying. However, due to the large difference in CTE between the $La_2Zr_2O_7$ top coating (~$9.0 \times 10^{-6}$/°C) and the $Yb_2Si_2O_7$ intermediate coating (~$4.1 \times 10^{-6}$/°C), the $La_2Zr_2O_7$ top layer finally fails after 24 cycles (360 min) during 1500 °C flame thermal shock.

The design of each layer thickness of the T/EBCs coating system also needs to be considered. Thickness is a key process parameter for coatings. Firstly, the principle of thickness design should be considered to meet the functional requirements. For example, the thickness of the TBC thermal insulation layer should meet the requirements of reducing the surface temperature to the target temperature of the EBC layer. And the thickness of the EBC layer should meet the ability of water and oxygen corrosion resistance during the service time. In addition, the thickness of the bonding layer should not be completely oxidized or fail during the service period. Secondly, the effect of thickness of each layer on the thermal stress of the entire coating system should be considered. Choose the appropriate thickness to reduce thermal stress, thus extending service life. Last but not the least, economic benefits need to be considered. Under the premise of satisfying the conditions of usage, the types and costs of raw materials consumed should be as few as possible. Therefore, it is necessary to study the thickness variation of T/EBCs coating system.

The research of T/EBCs is mainly carried out through experimental methods and finite element analysis (FEM) methods [34,35]. Harder et al. [36] evaluated different surface and intermediate layer material combinations in the residual stress analysis of sprayed material deposition for BSAS/Mullite/Si coating system through using FEM analysis. The result showed that the residual stress state is significantly affected by the surface layers of different BSAS phases (Hexacelsian BSAS and Celsian BSAS). However, it is less affected by the composition of the mullite interlayer. Richards et al. [17,19] simulated the thermal stress of $Yb_2SiO_5$/Mullite/Si EBCs fabricated by high-power and low-power APS after annealing at 1300 °C. The mechanism of crack bifurcation phenomenon in low power is explained by using parameters such as energy release rate (ERR), crack depth and phase angle. Heveran et al. [37] simulated the relationship between the thermal stress of TBC-EBCs (YSZ-Mullite) coating system and the deposition stress of sprayed material. The results indicated that the surface crack energy reduces the stress level in the TBC layer, but it induces a stress concentration near the crack tip. Nowadays, the numerical simulation technology of T/EBCs has achieved some valuable results. However, when the T/EBCs are applied under service conditions (such as thermal shock, water-oxygen corrosion, particle erosion, etc.), numerical studies are still limited and in increasing demand.

Recently, our group developed a TBC with a novel specific microstructure, namely embedded micro-agglomerated particle TBC (EMAP TBC) [38,39]. Compared with conventional 8YSZ TBC, EMAP 8YSZ TBC have more cracks and pores due to the presence of "embedded phase". As a result, it has better heat insulation and sintering resistance, and its elastic modulus is also low (~10 GPa). The flame thermal shock experiment showed

that the thermal cycle life of APS EMAP 8YSZ TBCs coating system is increased by more than four times at 1450 °C compared with the conventional coating system [38]. This because a moderate increase in microstructural features, such as cracks and pores, makes TBCs more compliant and strain-tolerant [40]. In order to enable EMAP TBC to be used at higher temperatures, gadolinium zirconate ($Gd_2Zr_2O_7$) material was investigated, and the corresponding EMAP $Gd_2Zr_2O_7$ TBC was developed. Since the thermal conductivity of $Gd_2Zr_2O_7$ material is lower than that of YSZ, and it also has good high-temperature phase stability below 1550 °C [41,42]. Therefore, $Gd_2Zr_2O_7$ is a potential TBC thermal insulation material that can be used at a higher temperature than YSZ.

This study applied the EMAP $Gd_2Zr_2O_7$ TBC layer to the $Yb_2Si_2O_7$/Si EBCs coating system through the finite element simulation of flame thermal shock at 1500 °C. And the influence of thickness change of each layer and composition change of EBC intermediate layer on temperature field and stress field during thermal shock was systematically researched. Firstly, the temperature and stress distributions of both EMAP $Gd_2Zr_2O_7$ T/EBCs (EMAP $Gd_2Zr_2O_7$/$Yb_2Si_2O_7$/Si) and conventional $Gd_2Zr_2O_7$ T/EBCs ($Gd_2Zr_2O_7$/$Yb_2Si_2O_7$/Si) coating system were compared on SiC substrate. Then, the thickness variation of EMAP $Gd_2Zr_2O_7$ T/EBCs coating system by both single variable and orthogonal experiment methods were performed, and the effect of thickness variation on temperature and stress distributions was performed. Finally, the $Yb_2Si_2O_7$ EBC intermediate layer doped with different content of $Yb_2SiO_5$ in the EMAP T/EBCs coating system was implemented. And the effects of $Yb_2SiO_5$ doping contents on temperature and stress distributions during thermal shock were also analyzed. These results are instructive to the application of novel EMAP T/EBCs coating system.

## 2. Experiments and Numerical Models

### 2.1. Coating Preparation

Figure 2 shows the coating spraying process. The $Gd_2Zr_2O_7$ powder and micro-aggregate $Gd_2Zr_2O_7$ powder are spherical with a particle size distribution of 20–80 μm and a purity of 99.9%, as shown in Figure 2a,b. In the spraying process of APS EMAP $Gd_2Zr_2O_7$ coating, two powder feeders were used, and the distance between them was 35 mm, as shown in Figure 2c. During the plasma dispersion process, except for the $Gd_2Zr_2O_7$ powder being fed into the front of the plasma jet through the powder feeder 1, the micro-agglomerated $Gd_2Zr_2O_7$ powder is also sent into the end of the plasma flame through the powder feeder 2. In this way, the APS EMAP $Gd_2Zr_2O_7$ coating with numerous pores and cracks is obtained, as shown in Figure 2d.

The conventional APS $Gd_2Zr_2O_7$ coatings and newly structured EMAP $Gd_2Zr_2O_7$ coatings were deposited on carbon steel substrates of Φ 25.4 mm × 3 mm and 25 mm × 5 mm × 1 mm, respectively. The APS system was equipped with an F4-MB plasma gun (Oerlikon Metco AG, Wohlen, Switzerland). The substrates were grit-blasted and cleaned before sprayed. Standard metallographic phase polishing was conducted on the cross-sectional of coatings. The powder morphology and microstructure of the coatings were characterized by using scanning electron microscopy (SEM; Hitachi Limited, Tokyo, Japan). The detailed APS operating parameters are listed in Table 1.

### 2.2. FE Model and Boundary Conditions

Figure 3 shows the two-dimensional finite element model for T/EBCs coating system. The SiC substrate is a coin-shaped cylinder of Φ 25.4 mm × 3 mm. The coating consists of three layers. Si is the bonding layer (BC). $Yb_2SiO_5$, $Yb_2Si_2O_7$ or $Yb_2SiO_5$ &$Yb_2Si_2O_7$ composite coatings are the EBC intermediate layer. $Gd_2Zr_2O_7$ and EMAP $Gd_2Zr_2O_7$ coatings are the TBC top layer. Half of the solid is used for modeling. Symmetry boundaries are set on the left side of the model and encastre boundaries are set at the bottom of the model. Other surfaces are not constrained. According to the actual flame thermal shock process [38], the initial temperature of the model is set to 25 °C, the upper surface is 1500 °C, and the lower surface is 1100 °C. The heating time, preservation time, and cooling time

are set to 50 s, 250 s and 120 s, respectively. The heat flux on the left and right boundary is 0 (adiabatic).

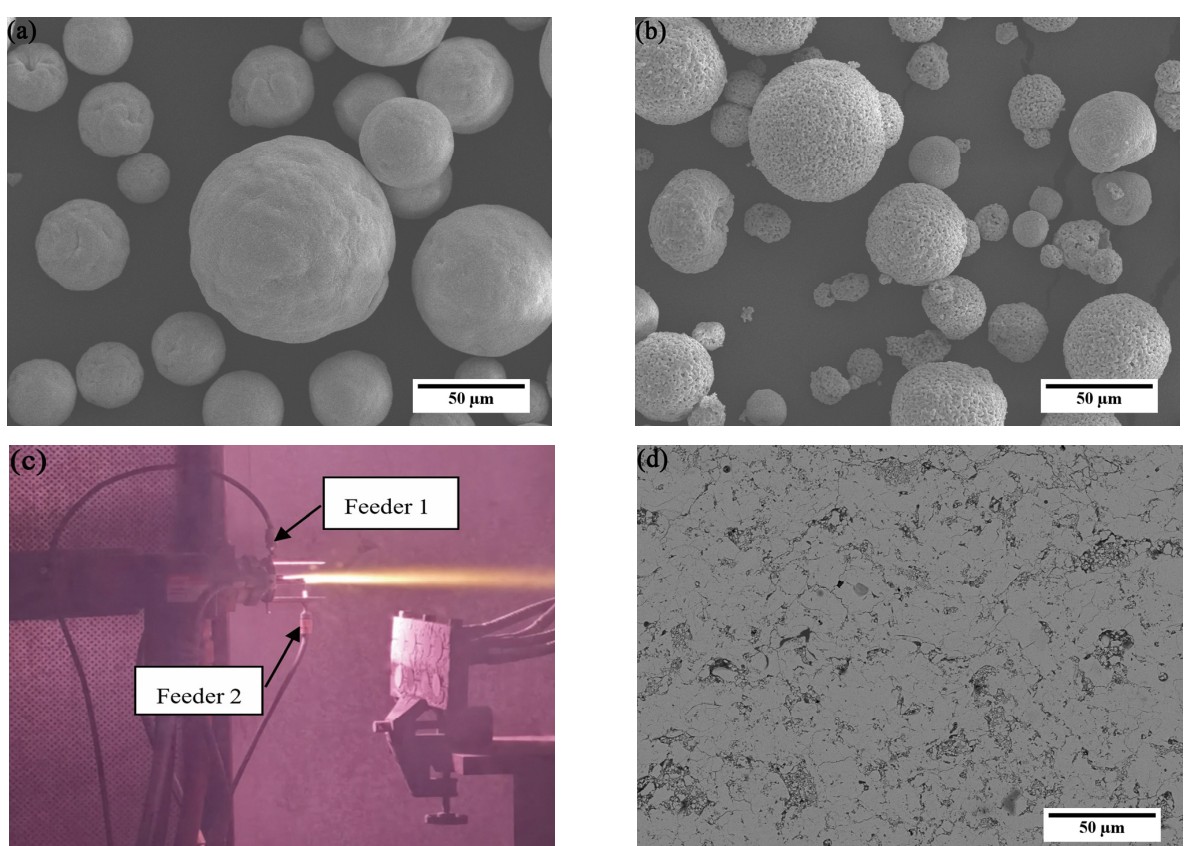

**Figure 2.** Coating spraying process: (**a**) Gd2Zr2O7 powder surface; (**b**) micro-aggregate Gd2Zr2O7 powder surface; (**c**) layout of equipment; (**d**) EMAP Gd2Zr2O7 coating.

**Table 1.** Operating parameters used for air plasma spray.

| APS Coatings | Power (kW) | Primary Ar (slm) | Secondary H$_2$ (slm) | Carrier Ar 1 (slm) | Carrier Ar 2 (slm) | Spray Distance (mm) |
|---|---|---|---|---|---|---|
| Gd$_2$Zr$_2$O$_7$ | 36 | 35 | 7 | 2 | N/A | 90 |
| EMAP Gd$_2$Zr$_2$O$_7$ | 36 | 35 | 7 | 2 | 4 | 90 |

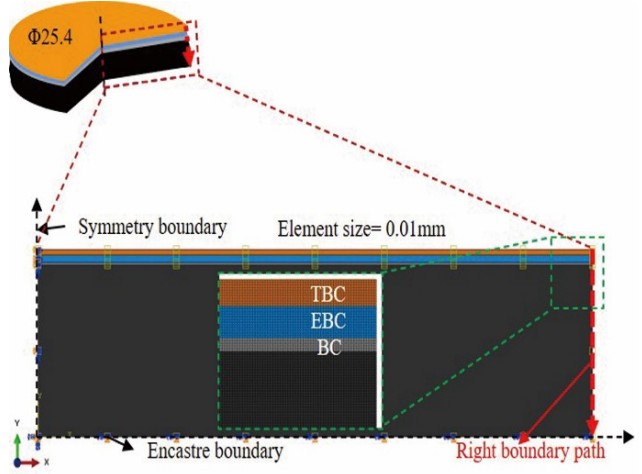

**Figure 3.** Finite element model for T/EBCs coating system.

The temperature field and stress field of thermal shock were simulated by the thermal sequential coupled model. All element types used in the temperature field model were 4-node linear heat transfer quadrilaterals (DC2D4). FE model is often treated with a two-dimensional plane strain case in the simulation study of TBCs [43–46]. Therefore, all element types used for the stress field model were 4-node bilinear plane strain quadrilaterals, reduced integration, hourglass control (CPE4R) in this study. After calculating the results of the temperature field model, then substituting it into the stress field model as a predefined field, the thermal stress distributions of the coating system at different times was obtained. To simplify the calculation, all element sizes were set to 0.01 mm. Finite element numerical simulations were performed using the commercial software ABAQUS 2021.

## 3. Material Parameters

### 3.1. Density, Young's Modulus, and Poisson's Ratio

Figure 4 shows the density, Young's modulus, and Poisson's ratio of the used materials. The density, Young's modulus and Poisson's ratio of the SiC substrate (SiC_Sub) are parameters from bulk materials [47]. The density of Si is taken from the bulk material [48]. The Young's modulus [49] and Poisson's ratio [20] of Si are parameters of coatings. The density [50,51], Young's modulus [21] and Poisson's ratio [52,53] of $Yb_2SiO_5$ and $Yb_2Si_2O_7$ are parameters of coatings. The density, Young's modulus and Poisson's ratio of 0.3 mol $Yb_2SiO_5$ & 0.7 mol $Yb_2Si_2O_7$ (0.3&0.7), 0.5 mol $Yb_2SiO_5$ & 0.5 mol $Yb_2Si_2O_7$ (0.5&0.5) and 0.7 mol $Yb_2SiO_5$ & 0.3 mol $Yb_2Si_2O_7$ (0.7&0.3) composite coatings are calculated according to the formulas from the literature [54,55] and the Reuss model [56], respectively. The densities of the $Gd_2Zr_2O_7$ (GZO) and EMAP $Gd_2Zr_2O_7$ (EMAP GZO) coatings are measured values in the laboratory (refer to Section 3.3 below for the detail measurement procedure). Leigh et al. [57] indicated that the thermally sprayed materials show the elastic modulus values that are 12%–78% of dense bulk materials, depending on the materials, spray processes, and post-treatments. Richards et al. [19,20] also take the Young's modulus of the APS coating as 50% of the bulk materials. Therefore, the Young's modulus of the conventional APS $Gd_2Zr_2O_7$ coating in this study is also taken half of the bulk materials from the literature [58]. And the Young's modulus of the EMAP $Gd_2Zr_2O_7$ coating is 10 GPa (refer to APS EMAP 8YSZ [38,39]). Both the Poisson's ratios of the $Gd_2Zr_2O_7$ and EMAP $Gd_2Zr_2O_7$ coatings are taken from bulk materials [58].

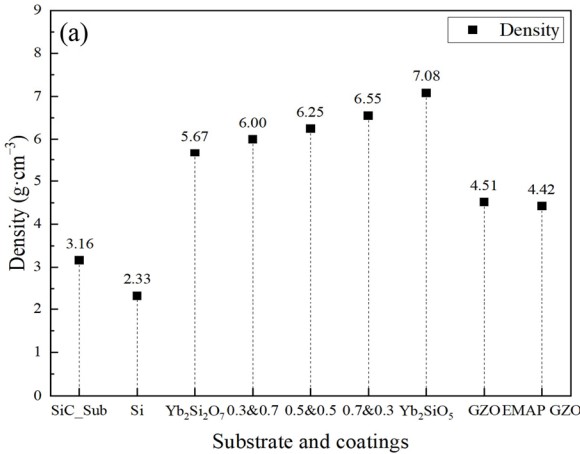
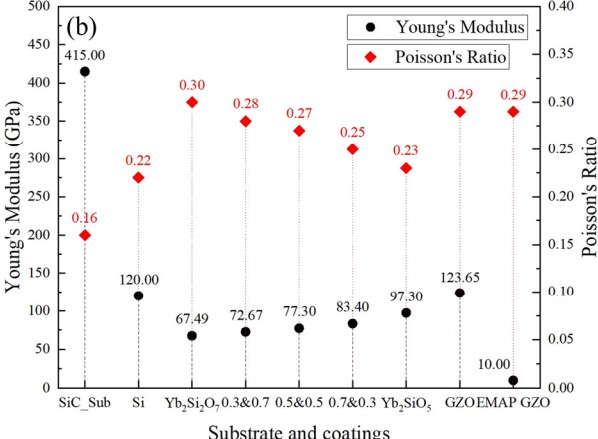

**Figure 4.** Distributions of density, Young's modulus and Poisson's ratio of materials: (**a**) density; (**b**) Young's modulus and Poisson's ratio.

### 3.2. Thermal Parameters

Figure 5 shows the thermal-related parameters of materials, and the values in Figure 5b are the arithmetic mean values of CTE of used materials in the range of 200–1400 °C of Figure 5a. The thermal expansion coefficient, thermal conductivity and specific heat of the SiC substrate are parameters bulk materials [47]. The thermal expansion coefficient and

specific heat of Si are also parameters from bulk materials [48], but its thermal conductivity is coating's parameter [59]. The thermal expansion coefficient, thermal conductivity and specific heat of $Yb_2SiO_5$ and $Yb_2Si_2O_7$ [21,50,51] are all parameters of coatings. The thermal expansion coefficient, thermal conductivity and specific heat parameters of 0.3 mol $Yb_2SiO_5$ & 0.7 mol $Yb_2Si_2O_7$ (0.3&0.7), 0.5 mol $Yb_2SiO_5$ & 0.5 mol $Yb_2Si_2O_7$ (0.5&0.5) and 0.7 mol $Yb_2SiO_5$ & 0.3 mol $Yb_2Si_2O_7$ (0.7&0.3) composite coatings are calculated according to the formulas from the Schapery model [60], the Rayleigh model [61] and the literature [55], respectively. The thermal expansion coefficient, thermal conductivity, and specific heat of $Gd_2Zr_2O_7$ (GZO) and EMAP $Gd_2Zr_2O_7$ (EMAP GZO) are the parameters of APS coatings measured in the laboratory (refer to Section 3.3 below for the detail measurement methods).

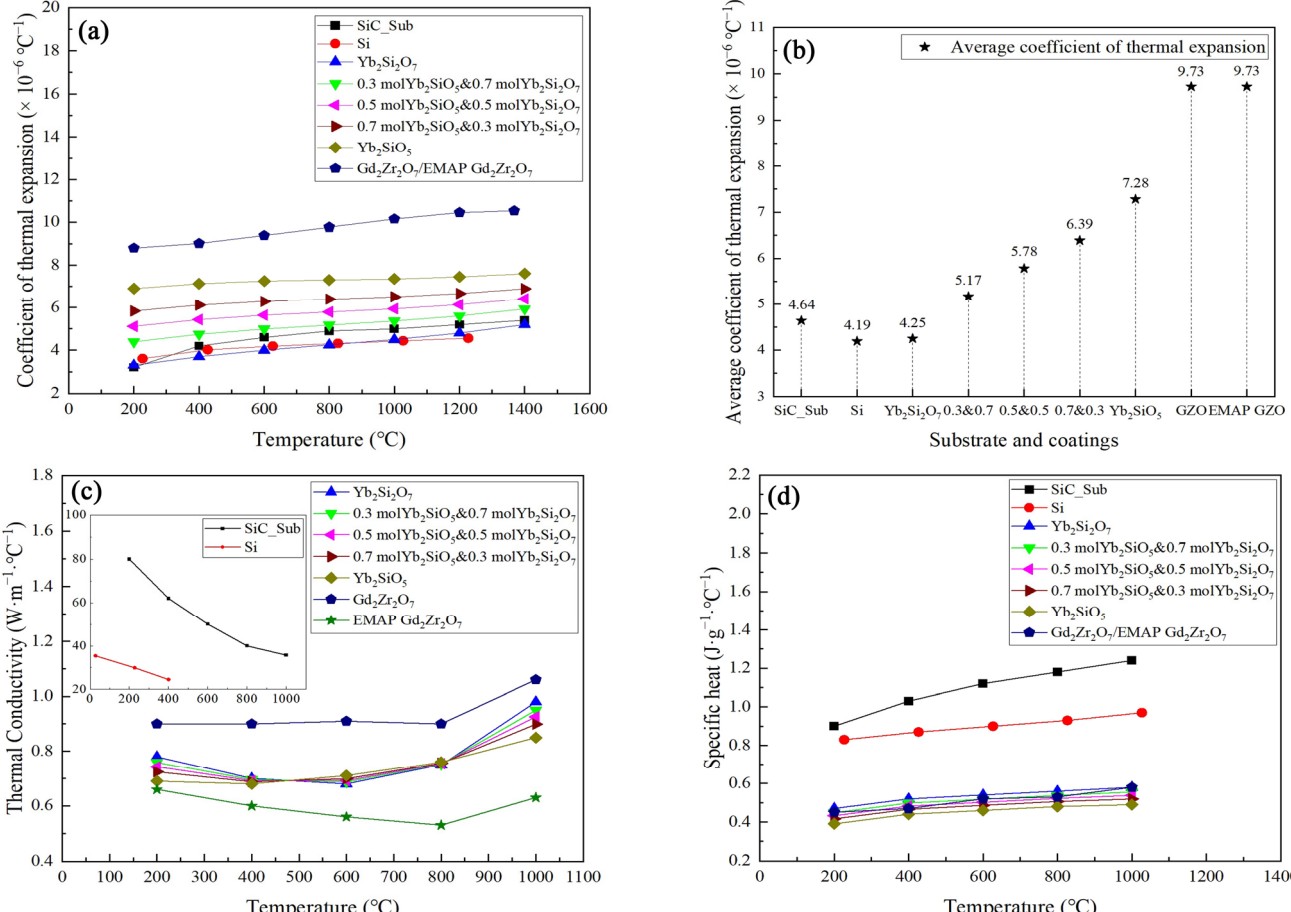

**Figure 5.** Distributions of thermally related parameters of materials: (**a**) CTE; (**b**) average CTE (200–1400 °C); (**c**) thermal conductivity; (**d**) specific heat.

### 3.3. Thermal Parameters Measurement

The thermophysical properties of APS $Gd_2Zr_2O_7$ and EMAP $Gd_2Zr_2O_7$ coatings measured in this study mainly include density, specific heat, thermal conductivity, and thermal expansion coefficient. The thermal conductivity (*k*) values were calculated using the following formula [62–65]:

$$k = c_p \cdot d \cdot \rho, \tag{1}$$

where $c_p$ is the heat capacity, *d* is the thermal diffusivity, and $\rho$ is the density. $c_p$ was determined by using a thermal analyzer and $\rho$ was measured by using the Archimedes method. Thermal diffusivity (*d*) was determined on coatings at 25–1000 °C by the laser flash method (model LFA 427, NETZSCH, Bavaria, Germany) using a thermal analyzer. Each temperature point was measured three times to ensure statistical consistency. The thermal expansion coefficient ($\alpha$) of the coatings was tested by means of a high temperature

dilatometer (model DIL 402E; Netzsch, Germany) from 25 °C to 1368 °C. The samples for these tests were cut from individual coatings measuring 25 mm × 5 mm × 1 mm.

The specific parameters of all materials used in the simulation are detailed in Tables A1–A3 in the Appendix A.

## 4. Results and Discussion

### 4.1. Comparison of EMAP T/EBCs with T/EBCs

#### 4.1.1. Temperature Distributions

Two different coating systems, T/EBCs coating system ($Gd_2Zr_2O_7$/$Yb_2Si_2O_7$/Si, GYS) and EMAP T/EBCs coating system (EMAP $Gd_2Zr_2O_7$/$Yb_2Si_2O_7$/Si, EGYS), are simulated. The thickness distributions of the first Si bonding layer and the second EBC layer is set to 50 μm and 120 μm according to the literature [20]. And the third TBC layer is set to 100 μm. Since this work focuses on the effects of thermo-mechanical parameters of different coatings on the temperature and stress distributions during thermal shock, the interface structure (such as roughness and oxides) and the complex feature structures (such as pores and cracks) inside the APS coating are neglected.

Figure 6 shows the temperature distributions along the right boundary path from coating surface to substrate. The thermal insulation capacity of the EMAP $Gd_2Zr_2O_7$ coating is better than that of the conventional $Gd_2Zr_2O_7$ coating (~50 °C lower at the TBC/EBC interface). This is mainly because the thermal conductivity of EMAP $Gd_2Zr_2O_7$ coating is about half that of conventional $Gd_2Zr_2O_7$ coating (as shown in Figure 5c). In addition, after the thermal insulation of the TBC and EBC layers, the temperature of the GYS and EGYS coating systems in the Si layer is approximately 1200 °C, which is much lower than the melting point of Si (~1416 °C) [14]. This means that the bonding layer of the two coating systems does not melt during the thermal shock.

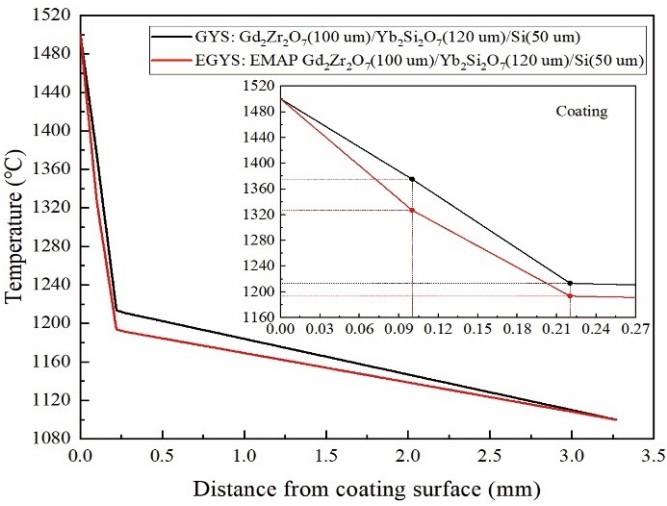

**Figure 6.** Temperature distributions along the right boundary path during heat preservation period (inset picture is a magnification of the coating section).

#### 4.1.2. Stress Distributions

Generally, the initiation of interfacial cracks mainly depends on the interfacial tensile stress in the thickness direction (y-axis), that is, the normal stress in the S22 direction [55]. For the convenience of description, the overall S22 tensile stress of the coating system (including the SiC substrate) is represented as S22_All, and S22_All_Max represents the maximum tensile stress of S22_All. It is found that the maximum value of S22_All_Max usually occurs at the beginning of the heat preservation or cooling stage. In order to compare the maximum tensile stress of different coating systems during the whole thermal shock process, the stress simulation results are all analyzed at the time when the maximum value of S22_All_Max is generated in this study.

Table 2 shows the maximum of S22 tensile stress (S22_Max) in each region of GYS and EGYS coating systems. The value of S22_All_Max for GYS coating system is 205.47 MPa, which is about 7 times higher than that of EGYS coating system (30.43 MPa), indicating that the GYS coating system is easier to failure due to thermal stress in the process of thermal shock. Furthermore, the S22_Max generation area of the GYS coating system is in the TBC layer. However, for the EGYS coating system, it is in the Si layer. Due to the bonding strength between the coating and the SiC substrate is different, the coating area is usually the initial place to crack [16–20]. From the results, it can be concluded that depositing 100 μm of APS conventional $Gd_2Zr_2O_7$ coating directly on the surface of $Yb_2Si_2O_7$ layer is not feasible. Because the TBC layer has high tensile stress in the GYS coating system, it may peel off during thermal shock. However, since the tensile stress of the EGYS coating system is much lower, applying the EMAP $Gd_2Zr_2O_7$ coating to the $Yb_2Si_2O_7$ layer by using an improved APS process appears to make it usable under thermal shock.

**Table 2.** S22_Max in each region of GYS and EGYS coating systems.

| Region | GYS (MPa) | EGYS (MPa) |
|---|---|---|
| SiC Substrate | 102.89 | 29.65 |
| Si layer | 100.07 | 30.43 |
| $Yb_2Si_2O_7$ layer | 92.17 | 30.01 |
| TBC layer | 205.47 | 10.84 |

Figure 7 shows the stress distributions from coating surface to substrate on the right boundary path. It shows that both the GYS and EGYS coating systems have tensile stress and compressive stress on the boundary path. In addition, the stress distributions of GYS coating system fluctuates more than that of EGYS. Furthermore, the value of S22_Max of the GYS coating systems (~200 MPa) is much larger than that of the EGYS coating system (~0 MPa).

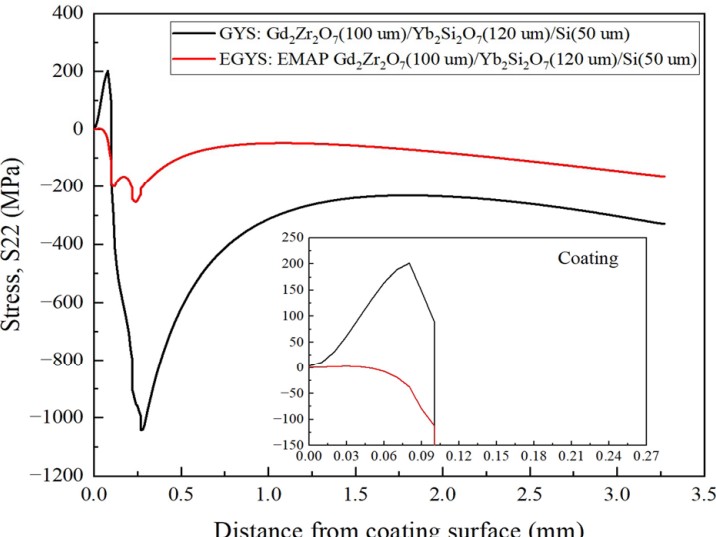

**Figure 7.** S22 distributions along right boundary path.

The different stress distributions of GYS and EGYS coating systems are mainly because the different of Young's modulus in the TBC layer. The Young's modulus is closely related to the thermal stress. It has been reported that the coating stress ($\sigma_c$) on SiC substrates is divided into three components [13,66,67]:

$$\sigma_c = \sigma_a + \sigma_g + \sigma_t, \tag{2}$$

where $\sigma_a$ is the aging stress; $\sigma_g$ is the growth stress; and $\sigma_t$ is the CTE mismatch stress. Aging stress is the stress due to changes in the physical, mechanical, and chemical properties of a coating caused by thermal exposure. Factors that cause these changes include phase transformation, sintering, oxidation, and chemical reactions. Aging stress can be minimized by selecting coating materials that remain phase stable, resist sintering, have low oxygen permeability, and are chemically compatible with the substrate and other layers during thermal cycling. Growth stress is the stress developed during coating deposition. Key parameters affecting APS growth stress include substrate temperature, plasma power, plasma gas, powder feed rate, powder carrier gas, spray spacing distance, powder particle size, and powder shape [13]. The CTE mismatch stress is the stress caused by the different CTEs of coating and substrate during heating and cooling stages, and the calculation formula is as follows [13,21,22]:

$$\sigma_t = \frac{(\alpha_c - \alpha_{sic})\Delta T E_c}{(1 - v_c)}, \tag{3}$$

where $\alpha_c$ and $\alpha_{sic}$ are the CTEs of the coating and SiC substrate, respectively; $E_c$ is the Young's modulus of the coating; $v_c$ is the Poisson's ratio of the coating; $\Delta T$ is the difference between the service temperature and the initial temperature. As can be seen from Equation (3), when the service temperature is determined, the thermal mismatch stress can be minimized by selecting coating materials with CTE closely matching with the substrate, low Young's modulus and small Poisson's ratio.

According to the above thermal shock simulation results of conventional T/EBCs (GYS) and EMAP T/EBCs (EGYS) coating systems, EMAP T/EBCs coating system has the characteristics of low thermal stress and strong thermal insulation ability. Therefore, it is a potential T/EBCs coating system for long-term use at 1500 °C. On the one hand, the EMAP $Gd_2Zr_2O_7$ coating is a porous structure with many "embedded phases", which can improve the sintering resistance and thermal insulation properties of the coating. In other words, reducing the aging stress of the coating. On the other hand, the porous structure of the "embedded phase" reduces the Young's modulus of the EMAP $Gd_2Zr_2O_7$ coating, thus reducing the CTE mismatch stress of the coating (Equation (3)). Hence, a three-layer coating design of EMAP T/EBCs for SiC-based ceramic substrates is proposed, as shown in Figure 8. Firstly, a thermal insulation TBC layer with low Young's modulus is deposited by APS for top layer, which is composed of EMAP TBC materials such as APS EMAP $Gd_2Zr_2O_7$. Secondly, an intermediate layer is made by APS rare earth pyro-silicate materials, such as APS $Yb_2Si_2O_7$, which are resistant to water and oxygen corrosion and the CTE matches the substrate. Thirdly, an anti-oxidative sacrificial bonding layer is needed at the bottom, which is prepared by CVD SiC, APS Si or the doped coatings of Si and refractory materials.

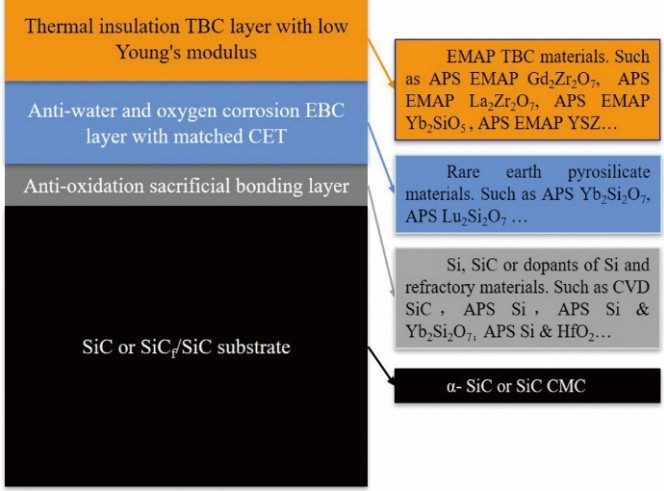

**Figure 8.** Three-layer coating design of EMAP T/EBCs coating system.

### 4.2. EMAP T/EBCs Thickness Design

#### 4.2.1. Single Variable Method

Figure 9 shows the temperature distributions of EGYS coating system at different thicknesses of EMAP $Gd_2Zr_2O_7$ top layer (h = 50, 80, 100, 150, 200, and 250 μm) during heat preservation period, in which the thicknesses of the Si layer and $Yb_2Si_2O_7$ layer were kept unchanged at 50 μm and 120 μm, respectively. Since the powder supplier recommend a long-term service temperature of 1350 °C for the $Yb_2Si_2O_7$ coating (refer to Metco 6157 specification), the EMAP $Gd_2Zr_2O_7$ insulation layer should not be less than 80 μm in order to reduce the temperature of the $Yb_2Si_2O_7$ layer from 1500 °C to below 1350 °C from the simulation results.

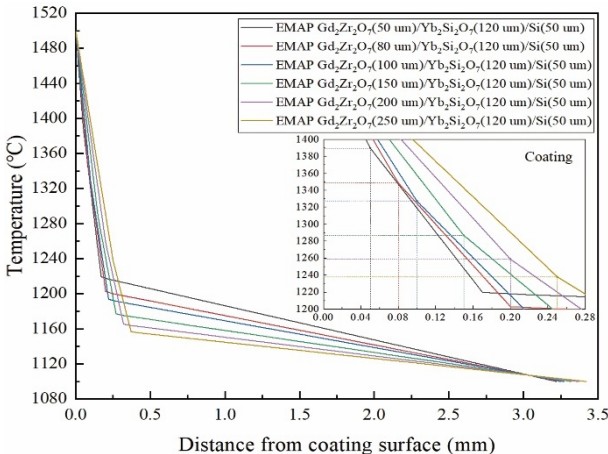

**Figure 9.** Temperature distributions of EGYS coating system at different thicknesses of EMAP $Gd_2Zr_2O_7$ layer (h = 50, 80, 100, 150, 200, and 250 μm) during heat preservation period.

Figure 10 shows the S22 stress distributions from surface to substrate along the right boundary path at different EMAP $Gd_2Zr_2O_7$ layer thicknesses. As the thickness of the EMAP $Gd_2Zr_2O_7$ layer increases from 50 μm to 250 μm, the S22_Max along the boundary path increases gradually from 0.75 MPa to 4.13 MPa. This may be because the fact that the average CTE of $Gd_2Zr_2O_7$ is different from that of $Yb_2Si_2O_7$ layer, Si layer and SiC substrate. The former is $9.73 \times 10^{-6}/°C$, while the latter are $4.25 \times 10^{-6}/°C$, $4.19 \times 10^{-6}/°C$ and $4.64 \times 10^{-6}/°C$, respectively (as shown in Figure 5b). However, due to the Young's modulus of the EMAP $Gd_2Zr_2O_7$ layer is small, the tensile stress at the boundary of the coating system is still low even though the thickness increases a lot.

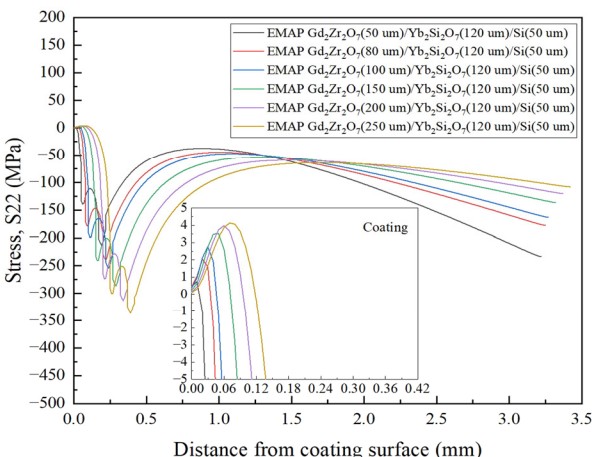

**Figure 10.** S22 stress distributions along right boundary path from surface to substrate at different EMAP $Gd_2Zr_2O_7$ layer thicknesses (h = 50, 80, 100, 150, 200, and 250 μm).

In a word, when the thicknesses of the Si layer and $Yb_2Si_2O_7$ layer is kept at 50 μm and 120 μm, and the thickness of the EMAP $Gd_2Zr_2O_7$ layer is changed at 50, 80, 100, 150, 200, and 250 μm, respectively, from the perspective of thermal insulation requirements, the thickness of the EMAP $Gd_2Zr_2O_7$ layer should not be less than 80 μm to ensure the temperature of the $Yb_2Si_2O_7$ layer is lower than 1350 °C during flame thermal shock at 1500 °C.

### 4.2.2. Orthogonal Experiment Method

In the EMAP $Gd_2Zr_2O_7$ T/EBCs coating system, in addition to considering the influence of the thickness of the EMAP $Gd_2Zr_2O_7$ layer on the thermal shocking stress, the thickness of the $Yb_2Si_2O_7$ and Si layers should also be considered. Based on the literature review of EBCs coating systems, it is found that the thickness of each layer mainly varies from 50 to 150 μm [14,16–21]. For ease of stress distributions description, S22_Max in the substrate region is denoted as S22_Sub_Max, and S22_Max in the coating is denoted as S22_Coat_Max. Similarly, the value of S22_Max in the Si layer can be denoted as S22_Si_Max, and the value of S22_Max in the corresponding $Yb_2Si_2O_7$ layer and EMAP $Gd_2Zr_2O_7$ can be denoted as S22_Ybds_Max and S22_Egzo_Max, respectively. Therefore, the orthogonal table of L-9 ($3^3$) can be used to count the S22_Max value of each layer with different thickness combinations during thermal shock, as is shown in the Table 3. It includes three factors: the thickness of the Si layer, the thickness of the $Yb_2Si_2O_7$ layer and the thickness of the EMAP $Gd_2Zr_2O_7$ layer. And the three levels of the table are 50, 100 and 150 μm.

**Table 3.** Values of S22_Max in each region of EGYS coating system in orthogonal experiment.

| Trial | Si (μm) | $Yb_2Si_2O_7$ (μm) | EMAP $Gd_2Zr_2O_7$ (μm) | S22_Sub_Max (MPa) | S22_Si_Max (MPa) | S22_Ybds_Max (MPa) | S22_Egzo_Max (MPa) |
|---|---|---|---|---|---|---|---|
| T1 | 50 | 50 | 50 | 36.75 | 30.19 | 28.13 | 9.98 |
| T2 | 50 | 100 | 100 | 30.10 | 31.00 | 30.28 | 11.55 |
| T3 | 50 | 150 | 150 | 31.14 | 31.32 | 30.89 | 12.16 |
| T4 | 100 | 50 | 100 | 27.18 | 30.82 | 25.50 | 13.00 |
| T5 | 100 | 100 | 150 | 29.29 | 31.33 | 29.04 | 13.16 |
| T6 | 100 | 150 | 50 | 26.82 | 23.82 | 22.71 | 8.05 |
| T7 | 150 | 50 | 150 | 25.38 | 30.10 | 22.91 | 14.21 |
| T8 | 150 | 100 | 50 | 29.75 | 24.79 | 22.77 | 8.15 |
| T9 | 150 | 150 | 100 | 23.63 | 28.10 | 26.94 | 9.53 |

Table 3 shows that the values of S22_All_Max is equal to S22_Sub_Max or S22_Si_Max. That is, the S22_All_Max of EMAP $Gd_2Zr_2O_7$ T/EBCs coating system is generated in the position of SiC substrate or Si layer. Furthermore, the S22_Max in the coating of the experimental group in descending order is: S22_Si_Max > S22_Ybds_Max > S22_Egzo_Max. Since the maximum S22 tensile stress of the coatings in the orthogonal experimental group is all in Si layer, that is, S22_Coat_Max is equal to S22_Si_Max. Therefore, reducing S22_Si_Max may be of great value in improving the durability of the coating system.

Figure 11 shows the stress trend of S22_Si_Max as a function of thickness of each layer in EGYS coating system. It shows that as the thickness of the Si layer and the $Yb_2Si_2O_7$ layer increases, the S22_Si_Max gradually decreases. However, as the thickness of the EMAP $Gd_2Zr_2O_7$ coating increases, the S22_Si_Max gradually increases. Therefore, in the EGYS coating system, it is better to increase the thickness of Si and $Yb_2Si_2O_7$ layers and decrease the thickness of EMAP $Gd_2Zr_2O_7$ layer. Furthermore, it also shows the thickness change of EMAP $Gd_2Zr_2O_7$ coating has the most significant influence on S22_Si_Max, followed by the thickness change of Si layer and $Yb_2Si_2O_7$ layer. This conclusion can be explained from the perspective of reducing CTE mismatch stress. Under the premise of constant temperature difference ($\Delta T$) and Poisson's ratio ($v_c$), the lower the Young's modulus of coating and the lower CTE difference between the substrate and the coating, the smaller

the CTE mismatch stress in the coating (Equation (3)). Compared with the Si layer and the EMAP $Gd_2Zr_2O_7$ layer, the average CTE of the $Yb_2Si_2O_7$ layer is closer to that of the substrate (Figure 5b), and the Young's modulus is also lower than that of Si, so the variation of its thickness has little effect on the CTE mismatch stress. For EMAP $Gd_2Zr_2O_7$ layer, although its Young's modulus is very low, the average CTE is much higher than that of the substrate, so the influence of the thickness change on the thermal stress is still dominant.

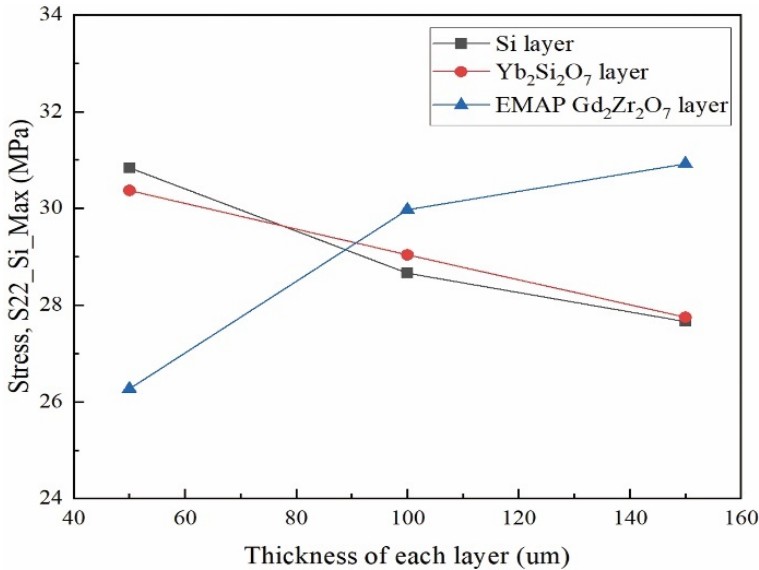

**Figure 11.** Stress trend of S22_Si_Max as a function of thickness of each layer in EGYS coating system.

Figure 12 shows the S22 stress distributions from surface to substrate along the right boundary path in the orthogonal experiments. With the variation of the thickness in the orthogonal experimental group, the S22_Max of the boundary path does not change much (<4 MPa). However, it can still be divided into three regions according to the thickness of the EMAP $Gd_2Zr_2O_7$ layer. This also indicates that the change of the thickness of EMAP $Gd_2Zr_2O_7$ layer has the most significant effect on S22 tensile stress at the boundary, while the Si layer and $Yb_2Si_2O_7$ layer have less effect.

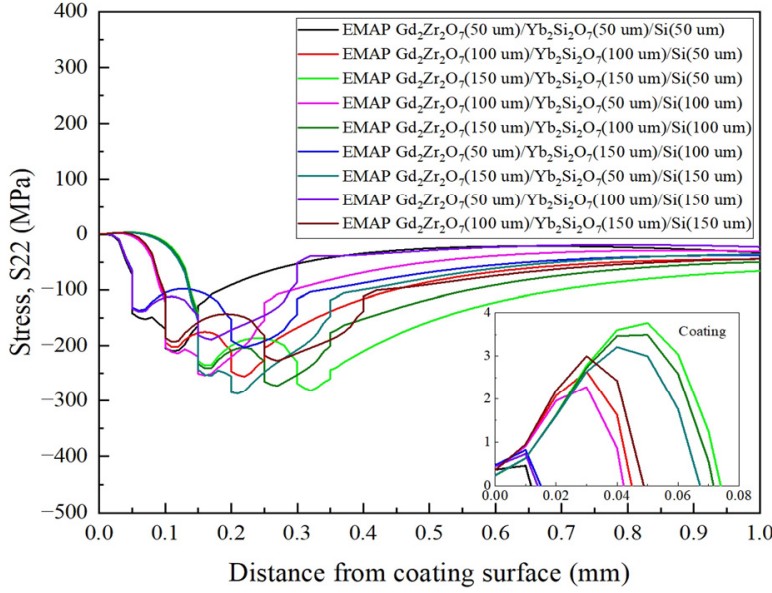

**Figure 12.** S22 stress distributions from surface to substrate along the right boundary path in the orthogonal experiments.

The analysis of the above orthogonal experimental results shows that when the thickness of each layer varies between 50, 100 and 150 μm in the EGYS coating system, from the perspective of reducing S22_Coat_Max/S22_Si_Max, the thickness of Si layer and $Yb_2Si_2O_7$ layer should be set as large as possible (i.e., 150 μm), while the thickness of EMAP $Gd_2Zr_2O_7$ layer is better to set as small as possible (i.e., 50 μm). Moreover, the thickness variation of the EMAP $Gd_2Zr_2O_7$ layer has the most significant effect on the S22 tensile stress, followed by the Si layer, and then the $Yb_2Si_2O_7$ layer. Therefore, considering the service temperature of the $Yb_2Si_2O_7$ layer (~1350 °C) and the experimental results of the single variable method, it is recommended that the thickness of the EMAP $Gd_2Zr_2O_7$ layer should be set to about 100 μm. While the thickness of the Si layer and the $Yb_2Si_2O_7$ layer can be set as large as possible to reduce the tensile stress.

### 4.3. Effect of $Yb_2SiO_5$ Doping

#### 4.3.1. Response of the Temperature Field

Figure 13 shows the effect of content of $Yb_2SiO_5$ on temperature distributions in EMAP $Gd_2Zr_2O_7$/$Yb_2Si_2O_7$/Si coating system. The thickness of Si layer and EMAP $Gd_2Zr_2O_7$ TBC layer are kept no change at 50 μm and 100 μm, respectively, and the thickness of EBC intermediate layer is also kept no change at 120 μm. It shows that under the same simulation conditions, with the increase in $Yb_2SiO_5$ content in the interlayer of the coating system, the thermal insulation ability of the EMAP $Gd_2Zr_2O_7$ layer decreases slowly. However, the thermal insulation effect of the EBC intermediate layer increases slightly. This may be because the high-temperature thermal conductivity of the $Yb_2SiO_5$ layer is a little lower than that of the $Yb_2Si_2O_7$ layer (Figure 5c).

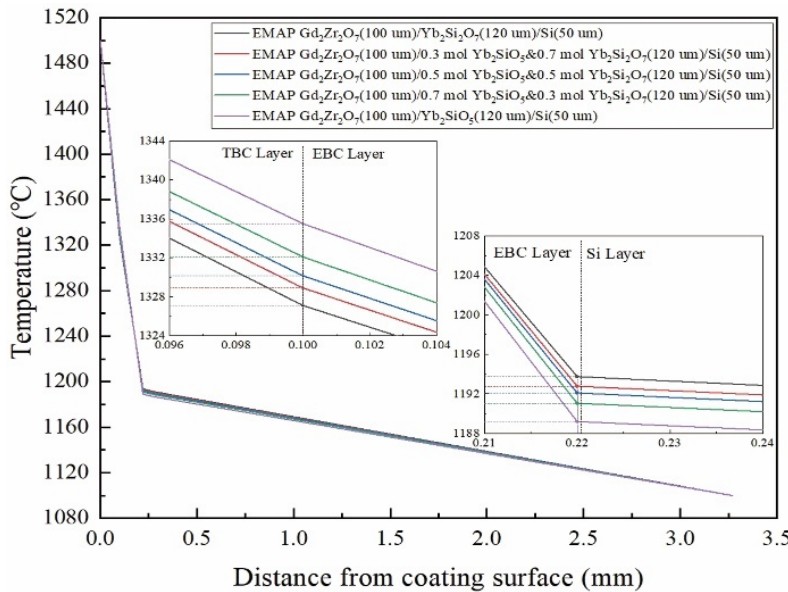

**Figure 13.** Effect of $Yb_2SiO_5$ content on temperature distributions during heat preservation period.

#### 4.3.2. Response of Stress Field

Figure 14 shows the effect of different $Yb_2SiO_5$ contents on the stress distributions. With the increase in the content of $Yb_2SiO_5$ in the interlayer, both the maximum S22 tensile stress and compressive stress increase. When the EBC interlayer is all $Yb_2SiO_5$, the maximum tensile stress of S22_Max is approximately doubled, which increases from 30.43 MPa to 61.64 MPa. Furthermore, with the increase in $Yb_2SiO_5$ content, the area of S22_All_Max (the bright red color) also gradually expands.

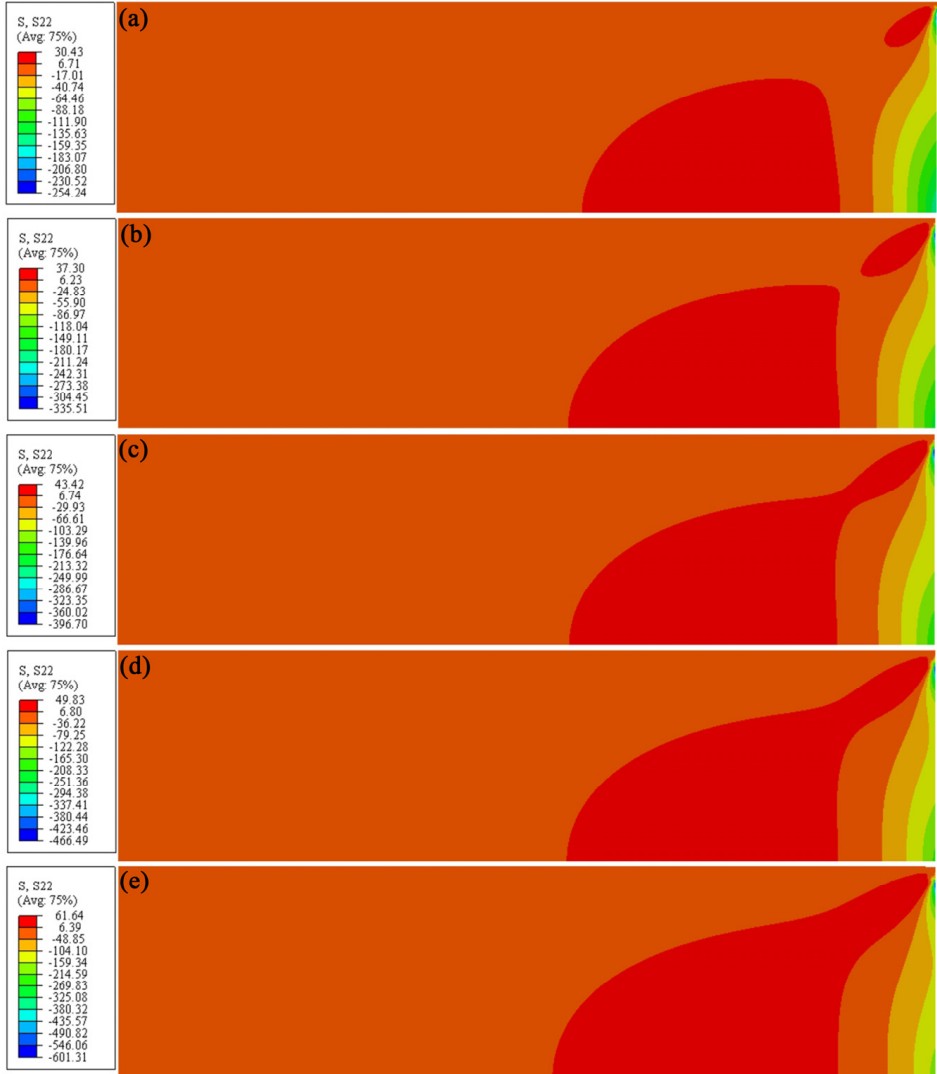

**Figure 14.** Effects of different Yb$_2$SiO$_5$ contents in the interlayer on stress distributions: (**a**) All Yb$_2$Si$_2$O$_7$; (**b**) 0.3 mol Yb$_2$SiO$_5$ & 0.7 mol Yb$_2$Si$_2$O$_7$; (**c**) 0.5 mol Yb$_2$SiO$_5$ & 0.5 mol Yb$_2$Si$_2$O$_7$; (**d**) 0.7 mol Yb$_2$SiO$_5$ & 0.3 mol Yb$_2$Si$_2$O$_7$; (**e**) All Yb$_2$SiO$_5$.

Table 4 is the statistical results of S22_Max of each region in Figure 14. It shows that the S22_All_Max generated region is in the SiC substrate or Si layer, and with the increase in Yb$_2$SiO$_5$ content in the interlayer, the S22_Max of other regions except the EMAP Gd$_2$Zr$_2$O$_7$ layer is gradually increasing. Furthermore, When the EBC interlayer is all Yb$_2$SiO$_5$, the value of S22_Coat_Max/S22_Si_Max increases from 30.43 MPa to 56.43 MPa. This indicates that the doping of Yb$_2$SiO$_5$ will exacerbate the tensile stress of the coating system during thermal shock, thereby increasing the risk of coating failure. However, the S22_Max of the EMAP Gd$_2$Zr$_2$O$_7$ layer showed a decreasing trend with the increase in the Yb$_2$SiO$_5$ content. This may be due to the increased CTE in the EBC interlayer layer of the coating system.

Figure 15 shows the stress distributions results of the right boundary of each model in Figure 14. It shows that with the increase in Yb$_2$SiO$_5$ content, the S22_Max of boundary is slowly decreasing. Generally, the S22 tensile stress on the coating boundary is very small (<3 MPa). This means that the change of Yb$_2$SiO$_5$ content in the intermediate layer has little effect on the boundary stress distributions. However, with the increase in Yb$_2$SiO$_5$ content, the compressive stress at the interface between substrate and Si layer of the coating system increases greatly. The effect of this phenomenon on coatings is uncertain.

**Table 4.** S22_Max of each region with different $Yb_2SiO_5$ contents.

| Region | $Yb_2Si_2O_7$ (MPa) | 0.3 mol $Yb_2SiO_5$ & 0.7 mol $Yb_2Si_2O_7$ (MPa) | 0.5 mol $Yb_2SiO_5$ & 0.5 mol $Yb_2Si_2O_7$ (MPa) | 0.7 mol $Yb_2SiO_5$ & 0.3 mol $Yb_2Si_2O_7$ (MPa) | $Yb_2SiO_5$ (MPa) |
|---|---|---|---|---|---|
| Substrate | 29.65 | 37.30 | 43.42 | 49.83 | 61.64 |
| Si layer | 30.43 | 36.95 | 42.03 | 47.20 | 56.43 |
| EBC interlayer | 30.01 | 33.06 | 35.40 | 37.95 | 42.76 |
| EMAP $Gd_2Zr_2O_7$ layer | 10.84 | 9.08 | 8.01 | 7.12 | 5.87 |

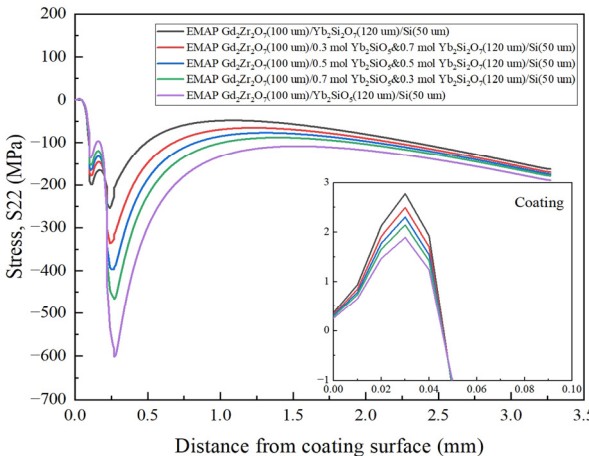

**Figure 15.** Effect of $Yb_2SiO_5$ content on the S22 stress distributions along the right boundary path.

To sum up, the doping of $Yb_2SiO_5$ in the $Yb_2Si_2O_7$ intermediate of the EMAP $Gd_2Zr_2O_7$/ $Yb_2Si_2O_7$/Si coating system increases the maximum of S22 tensile stress. Moreover, when the EBC interlayer is all $Yb_2SiO_5$ coating, the S22_Max value is about doubled. Therefore, doping $Yb_2SiO_5$ in the $Yb_2Si_2O_7$ intermediate layer is not recommended from the point of reducing thermal shock stress. However, if the bonding strength between the Si layer and the composite layer is good enough, in order to improve the water-oxygen corrosion resistance of the coating system, it is advisable to dope an appropriate amount of $Yb_2SiO_5$ in the $Yb_2Si_2O_7$ layer. But this needs to be verified through a series of related experiments.

## 5. Conclusions

In this study, the finite element simulation of flame thermal shock at 1500 °C was carried out for the novel structural EMAP $Gd_2Zr_2O_7$ T/EBCs (EMAP $Gd_2Zr_2O_7$/$Yb_2Si_2O_7$/Si) coating system. Firstly, the novel coating system was compared with the conventional $Gd_2Zr_2O_7$/$Yb_2Si_2O_7$/Si coating system on the SiC substrate, indicating that it is feasible to use EMAP $Gd_2Zr_2O_7$ TBC as thermal insulation layer on $Yb_2Si_2O_7$/Si coating system. Based on that, a three-layer coating design of EMAP T/EBCs for SiC-based ceramic substrates was proposed. Secondly, the influence of the thickness variation of each layer on the temperature and stress distributions in the novel coating system was analyzed by both single variable and orthogonal experiment methods, which provided a reference for the thickness design for this coating system. Finally, the $Yb_2Si_2O_7$ interlayer with doping different contents of $Yb_2SiO_5$ was carried out, and the results of different $Yb_2SiO_5$ content on the temperature and stress distributions were revealed. The major conclusions can be summarized as follows:

(1) Compared with conventional $Gd_2Zr_2O_7$ TBC layer, APS EMAP $Gd_2Zr_2O_7$ TBC layer has a lower Young's modulus due to the introduction of more cracks and pores, which greatly reduces the thermal tensile stress in the T/EBCs coating system. According to the simulation results of conventional $Gd_2Zr_2O_7$/$Yb_2Si_2O_7$/Si coating system, the maximum tensile stress is 205.47 MPa in the $Gd_2Zr_2O_7$ layer. However, the maximum tensile stress of

EMAP $Gd_2Zr_2O_7/Yb_2Si_2O_7/Si$ coating system is only 30.43 MPa, and it is generated in the Si layer.

(2) When the thickness of each layer of the EMAP $Gd_2Zr_2O_7$ T/EBCs coating system varies in the range of 50–150 μm, considering the long-term service requirement of $Yb_2Si_2O_7$ and the requirement of reducing the maximum tensile stress of coating system, the thickness of EMAP $Gd_2Zr_2O_7$ coating layer is recommended to be set at about 100 μm. However, the thickness of $Yb_2Si_2O_7$ and Si layers can be set as large as necessary.

(3) With the increase in $Yb_2SiO_5$ content in the $Yb_2Si_2O_7$ intermediate layer of the EMAP $Gd_2Zr_2O_7$ T/EBCs coating system, the maximum tensile stress and the corresponding maximum tensile stress area increase gradually. When the middle layer is replaced by $Yb_2SiO_5$, the maximum S22 tensile stress in the coating system increases from 30.43 MPa to 56.43 MPa. This indicates that the increase in $Yb_2SiO_5$ doping in the $Yb_2Si_2O_7$ intermediate layer aggravates the spalling failure risk of the novel coating system.

**Author Contributions:** Conceptualization, K.L. and W.W.; methodology, K.L.; software, K.L.; validation, K.L., W.W., T.Y. and Y.L.; formal analysis, K.L.; investigation, T.Y.; resources, K.L.; data curation, Y.L.; writing—original draft preparation, K.L.; writing—review and editing, W.W.; visualization, K.L.; supervision, W.W.; project administration, W.W.; funding acquisition, W.W. All authors have read and agreed to the published version of the manuscript.

**Funding:** This research was funded by the National Natural Science Foundation of China (No. 52175136, No. 52130511), Science Center for Gas Turbine Project (No. P2021-A-IV-002-002), The National High Technology Research and Development Program of China (No. 2021YFB3702200), Key Research and Development Projects in Anhui Province (No. 2022a05020004).

**Institutional Review Board Statement:** Not applicable.

**Informed Consent Statement:** Not applicable.

**Data Availability Statement:** The raw/processed data required to reproduce these findings cannot be shared at this time as the data also forms part of an ongoing study.

**Acknowledgments:** The author thanks the contribution of W.W. for reviewing and discussing this work; T.Y. for providing parameters information of $Gd_2Zr_2O_7$ and EMAP $Gd_2Zr_2O_7$ coatings, as well as for the SEM photographing of EMAP $Gd_2Zr_2O_7$ coatings; Y.L. for discussing of simulation results; Research Center of Analysis and Test of ECUST for the characterization of Hitachi S-3400N.

**Conflicts of Interest:** The authors declare no conflict of interest.

## Appendix A

**Table A1.** Thermophysical parameters of substrate and EBCs.

| Material | Temperature (°C) | Thermal Expansion Coefficient ($\times 10^{-6} \cdot °C^{-1}$) | Thermal Conductivity ($W \cdot m^{-1} \cdot °C^{-1}$) | Specific Heat ($J \cdot g^{-1} \cdot °C^{-1}$) | Elastic Modulus (Gpa) | Poisson's Ratio | Density ($g \cdot cm^{-3}$) |
|---|---|---|---|---|---|---|---|
| α-SiC | 20 | - | - | - | 415.00 | 0.16 | 3.16 |
| | 200 | 3.20 | 80.00 | 0.90 | - | - | - |
| | 400 | 4.20 | 62.00 | 1.03 | - | - | - |
| | 600 | 4.60 | 50.00 | 1.12 | - | - | - |
| | 800 | 4.90 | 40.00 | 1.18 | - | - | - |
| | 1000 | 5.00 | 35.70 | 1.24 | - | - | - |
| | 1200 | 5.20 | - | - | - | - | - |
| | 1400 | 5.40 | - | - | - | - | - |
| Si | 27 | - | 35.40 | - | 120 | 0.22 | 2.33 |
| | 227 | 3.61 | 29.90 | 0.83 | - | - | - |
| | 400 | - | 24.50 | - | - | - | - |
| | 427 | 4.02 | - | 0.87 | - | - | - |

**Table A1.** Thermophysical parameters of substrate and EBCs.

| Material | Temperature (°C) | Thermal Expansion Coefficient ($\times 10^{-6} \cdot {}^{\circ}C^{-1}$) | Thermal Conductivity ($W \cdot m^{-1} \cdot {}^{\circ}C^{-1}$) | Specific Heat ($J \cdot g^{-1} \cdot {}^{\circ}C^{-1}$) | Elastic Modulus (Gpa) | Poisson's Ratio | Density ($g \cdot cm^{-3}$) |
|---|---|---|---|---|---|---|---|
| | 627 | 4.19 | - | 0.90 | - | - | - |
| | 827 | 4.32 | - | 0.93 | - | - | - |
| | 1027 | 4.44 | - | 0.97 | - | - | - |
| | 1227 | 4.56 | - | - | - | - | - |
| $Yb_2SiO_5$ | 25 | - | - | - | 97.30 | 0.23 | 7.08 |
| | 200 | 6.90 | 0.69 | 0.39 | - | - | - |
| | 400 | 7.13 | 0.68 | 0.44 | - | - | - |
| | 600 | 7.25 | 0.71 | 0.46 | - | - | - |
| | 800 | 7.30 | 0.76 | 0.48 | - | - | - |
| | 1000 | 7.35 | 0.85 | 0.49 | - | - | - |
| | 1200 | 7.45 | - | - | - | - | - |
| | 1400 | 7.60 | - | - | - | - | - |
| $Yb_2Si_2O_7$ | 25 | - | - | - | 67.49 | 0.30 | 5.67 |
| | 200 | 3.30 | 0.78 | 0.47 | - | - | - |
| | 400 | 3.70 | 0.70 | 0.52 | - | - | - |
| | 600 | 4.00 | 0.68 | 0.54 | - | - | - |
| | 800 | 4.25 | 0.75 | 0.56 | - | - | - |
| | 1000 | 4.50 | 0.98 | 0.58 | - | - | - |
| | 1200 | 4.80 | - | - | - | - | - |
| | 1400 | 5.20 | - | - | - | - | - |

**Table A2.** Thermophysical parameters of ytterbium silicate composite coating.

| Material | Temperature (°C) | Thermal Expansion Coefficient ($\times 10^{-6} \cdot {}^{\circ}C^{-1}$) | Thermal Conductivity ($W \cdot m^{-1} \cdot {}^{\circ}C^{-1}$) | Specific Heat ($J \cdot g^{-1} \cdot {}^{\circ}C^{-1}$) | Elastic Modulus (Gpa) | Poisson's Ratio | Density ($g \cdot cm^{-3}$) |
|---|---|---|---|---|---|---|---|
| 0.3 mol $Yb_2SiO_5$ & 0.7 mol $Yb_2Si_2O_7$ | 25 | - | - | - | 72.67 | 0.28 | 6.00 |
| | 200 | 4.39 | 0.76 | 0.45 | - | - | - |
| | 400 | 4.74 | 0.70 | 0.50 | - | - | - |
| | 600 | 4.99 | 0.69 | 0.52 | - | - | - |
| | 800 | 5.18 | 0.75 | 0.54 | - | - | - |
| | 1000 | 5.37 | 0.95 | 0.56 | - | - | - |
| | 1200 | 5.61 | - | - | - | - | - |
| | 1400 | 5.93 | - | - | - | - | - |
| 0.5 mol $Yb_2SiO_5$ & 0.5 mol $Yb_2Si_2O_7$ | 25 | - | - | - | 77.30 | 0.27 | 6.25 |
| | 200 | 5.12 | 0.74 | 0.43 | - | - | - |
| | 400 | 5.43 | 0.69 | 0.48 | - | - | - |
| | 600 | 5.64 | 0.69 | 0.50 | - | - | - |
| | 800 | 5.79 | 0.75 | 0.52 | - | - | - |
| | 1000 | 5.94 | 0.93 | 0.54 | - | - | - |
| | 1200 | 6.14 | - | - | - | - | - |
| | 1400 | 6.41 | - | - | - | - | - |
| 0.7 mol $Yb_2SiO_5$ & 0.3 mol $Yb_2Si_2O_7$ | 25 | - | - | - | 83.40 | 0.25 | 6.55 |
| | 200 | 5.83 | 0.72 | 0.42 | - | - | - |
| | 400 | 6.12 | 0.69 | 0.47 | - | - | - |
| | 600 | 6.29 | 0.70 | 0.49 | - | - | - |
| | 800 | 6.40 | 0.76 | 0.51 | - | - | - |
| | 1000 | 6.51 | 0.90 | 0.52 | - | - | - |
| | 1200 | 6.67 | - | - | - | - | - |
| | 1400 | 6.89 | - | - | - | - | - |

**Table A3.** Thermophysical parameters of TBCs.

| Material | Temperature (°C) | Thermal Expansion Coefficient ($\times 10^{-6} \cdot °C^{-1}$) | Thermal Conductivity ($W \cdot m^{-1} \cdot °C^{-1}$) | Specific Heat ($J \cdot g^{-1} \cdot °C^{-1}$) | Elastic Modulus (Gpa) | Poisson's Ratio | Density ($g \cdot cm^{-3}$) |
|---|---|---|---|---|---|---|---|
| $Gd_2Zr_2O_7$ | 25 | - | - | - | 123.65 | 0.29 | 4.51 |
| | 200 | 8.80 | 0.90 | 0.45 | - | - | - |
| | 400 | 9.02 | 0.90 | 0.47 | - | - | - |
| | 600 | 9.39 | 0.91 | 0.52 | - | - | - |
| | 800 | 9.77 | 0.90 | 0.53 | - | - | - |
| | 1000 | 10.15 | 1.06 | 0.58 | - | - | - |
| | 1200 | 10.45 | - | - | - | - | - |
| | 1368 | 10.53 | - | - | - | - | - |
| EMAP $Gd_2Zr_2O_7$ | 25 | - | - | - | 10 | 0.29 | 4.42 |
| | 200 | 8.80 | 0.66 | 0.45 | - | - | - |
| | 400 | 9.02 | 0.60 | 0.47 | - | - | - |
| | 600 | 9.39 | 0.56 | 0.52 | - | - | - |
| | 800 | 9.77 | 0.53 | 0.53 | - | - | - |
| | 1000 | 10.15 | 0.63 | 0.58 | - | - | - |
| | 1200 | 10.45 | - | - | - | - | - |
| | 1368 | 10.53 | - | - | - | - | - |

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
