# Peer review of "Simulation of 1500 °C Thermal Shock for Novel Structural Thermal/Environmental Barrier Coatings System"

_coatings, doi:10.3390/coatings13010096_

Round 1

Reviewer 1 Report

Article review Simulation of 1500 °C thermal shock for novel structural thermal/environmental barrier coatings system”.

The article is devoted to the development of new thermal barrier coatings. In this study, an innovative coating system EMAP Gd2Zr2O7 T/EBCs (EMAT Gd2Zr2O7/Yb2Si2O7/Si) at a flame temperature of 1500 °C was simulated and systematically studied on SiC substrate. The results showed that the Gd2Zr2O7 T/EBCs EMAP coating system has much lower thermal voltage performance than the conventional Gd2Zr2O7/Yb2Si2O7/Si T/EBCs coating system. In the work, new interesting results were obtained, however, several questions and comments arose:

1 In Figures 3 and 4b, the inscriptions on the x-axis are poorly visible.

2 No data used for thermal conductivity according to formula 1 on page 7 - heat capacity (Cp), thermal diffusivity (d)

3 The paper does not show information about Yb2SiO5, Yb2Si2O7. X-ray patterns, cell parameters are not shown, the microstructure of the compounds is not shown. In the future, the distribution of these phases in a complex composite - ytterbium silicate composite is not considered.

Reviewer 2 Report

The study applied the EMAP Gd2Zr2O7 TBC layer to the Yb2Si2O7/Si EBCs coating system through finite element simulation of flame thermal shock at 1500 ℃. In addition, the influence of the change in thickness of each layer and the change in the composition of the intermediate layer.

The work is interesting and well structured. However, a specification about the application would be interesting. The author just quotes, however could insert a table with the required temperature and application. In this way, the work could be valued with the possible applications.

Other suggestions:

Change these keywords: Environmental barrier coatings; Thermal barrier coatings; Simulation of thermal shock, as they are in the title.
